# Vitamin D induces SIRT1 activation through K610 deacetylation in colon cancer

José Manuel García-Martínez[1], Ana Chocarro-Calvo[1], Javier Martínez-Useros[1,2], María Jesús Fernández-Aceñero[3], M Carmen Fiuza[4], José Cáceres-Rentero[1], Antonio De la Vieja[5,6], Antonio Barbáchano[6,7,8], Alberto Muñoz[6,7,8], María Jesús Larriba[6,7,8], Custodia García-Jiménez[1]*

[1]Area of Physiology, Faculty Health Sciences, University Rey Juan Carlos, Alcorcón, Madrid, Spain; [2]Translational Oncology Division, OncoHealth Institute, Health Research Institute-University Hospital Fundación Jiménez Díaz-Universidad Autónoma de Madrid, Madrid, Spain; [3]Department of Surgical Pathology, Hospital Clínico San Carlos, Madrid, Spain; [4]Department of Surgery, University Hospital Fundación Alcorcón-Universidad Rey Juan Carlos, Alcorcón, Madrid, Spain; [5]Unidad de Tumores Endocrinos (UFIEC), Instituto de Salud Carlos III, Majadahonda, Madrid, Spain; [6]CIBER de Cáncer, Instituto de Salud Carlos III, Madrid, Spain; [7]Instituto de Investigaciones Biomédicas "Alberto Sols", Consejo Superior de Investigaciones Científicas, Universidad Autónoma de Madrid, Madrid, Spain; [8]Instituto de Investigación Sanitaria del Hospital Universitario La Paz, Madrid, Spain

*For correspondence:
custodia.garcia@urjc.es

Competing interest: The authors declare that no competing interests exist.

**Abstract** Posttranslational modifications of epigenetic modifiers provide a flexible and timely mechanism for rapid adaptations to the dynamic environment of cancer cells. SIRT1 is an $NAD^+$-dependent epigenetic modifier whose activity is classically associated with healthy aging and longevity, but its function in cancer is not well understood. Here, we reveal that 1α,25-dihydroxyvitamin $D_3$ (1,25(OH)$_2D_3$, calcitriol), the active metabolite of vitamin D (VD), promotes SIRT1 activation through auto-deacetylation in human colon carcinoma cells, and identify lysine 610 as an essential driver of SIRT1 activity. Remarkably, our data show that the post-translational control of SIRT1 activity mediates the antiproliferative action of 1,25(OH)$_2D_3$. This effect is reproduced by the SIRT1 activator SRT1720, suggesting that SIRT1 activators may offer new therapeutic possibilities for colon cancer patients who are VD deficient or unresponsive. Moreover, this might be extrapolated to inflammation and other VD deficiency-associated and highly prevalent diseases in which SIRT1 plays a prominent role.

## eLife assessment

This study demonstrates that vitamin D-bound VDR increased the expression of SIRT1 and that vitamin D-bound VDR interacts with SIRT1 to cause auto-deacetylation on Lys610 and activation of SIRT1 catalytic activity. This is an **important** finding that is relevant to the actions of VDR on colorectal cancer. The data presented to support the presented conclusion are **convincing**.

## Introduction

Over the last decades the study of genetic lesions in tumour cells provided key insights into the deregulated signalling pathways that promote cell proliferation and invasiveness and suppress senescence. In addition, increasing evidence from epidemiological studies suggests a significant impact of non-genetic modifiable factors on the oncogenic process by affecting cell epigenetics and thus, gene expression and phenotype. The key metabolic sensor SIRT1 acts an $NAD^+$-dependent deacetylase and a critical post-translational modifier and epigenetic regulator. As such, SIRT1 promotes efficient energy utilisation and cellular defences in response to environmental challenges, whereas its dysregulation accelerates age-related diseases such as diabetes and cancer (*Bonkowski and Sinclair, 2016*).

Colorectal cancer (CRC) is the second most diagnosed malignancy in women, the third in men, and a leading cause of cancer-related deaths worldwide, with an incidence estimated to increase by 60% by 2030 (*Bray et al., 2018*; *The Cancer Genome Atlas Network, 2012*). Many observational/epidemiological studies suggest that vitamin D (VD) deficiency is a risk factor for developing and dying of cancer, particularly for CRC (*Feldman et al., 2014*; *Grant et al., 2022*; *Kim et al., 2023*, *Kim et al., 2021*), albeit data from supplementation studies in the human population are controversial. Confirmation of a clinically relevant anti-CRC effect of VD in well-designed prospective randomized studies is still pending (*Chandler et al., 2020*; *Henn et al., 2022*; *Manson et al., 2020*; *Muñoz and Grant, 2022*; *Song et al., 2021*).

VD in humans has two origins: synthesis in the skin by the action of solar ultraviolet radiation on 7-dehydrocholesterol and the diet. The active VD metabolite is 1α,25-dihydroxyvitamin $D_3$ (1,25(OH)$_2D_3$, calcitriol), which results from two consecutive hydroxylations of VD, the first in the liver and the second in the kidney or in many epithelial and immune cell types (*Bikle and Christakos, 2020*; *Feldman et al., 2014*). VD actions are mediated by 1,25(OH)$_2D_3$ binding to a member of the nuclear receptor superfamily of transcription factors, the VD receptor (VDR). Upon ligand binding, VDR regulates the expression of hundreds of target genes, many of which are involved in calcium and phosphate homeostasis, bone biology, immune response, metabolism, detoxification, and cell survival, proliferation and differentiation (*Bikle and Christakos, 2020*; *Carlberg and Muñoz, 2022a*; *Carlberg and Velleuer, 2022b*; *Feldman et al., 2014*; *Fernández-Barral et al., 2020*).

Consistent with a protective role of VD against CRC, the colonic epithelium expresses high levels of VDR. Many studies in CRC cells and associated fibroblasts and in experimental animals have shown a large series of VDR-mediated VD anti-CRC effects (*Carlberg and Muñoz, 2022a*; *Carlberg and Velleuer, 2022b*; *Feldman et al., 2014*; *Fernández-Barral et al., 2020*; *Ferrer-Mayorga et al., 2019*) In support, germline deletion of *Vdr* in mice with constitutively active Wnt/β-catenin signalling, as a main driver of CRC, results in increased intestinal tumour load (*Larriba et al., 2011*; *Zheng et al., 2012*).

Notably, certain evidence links SIRT1 and VD. First, decreased SIRT1 activity has been linked to the pathogenesis of CRC (*Ren et al., 2017*; *Strycharz et al., 2018*). However, using SIRT1 level as a marker of malignancy is controversial since both increased or decreased SIRT1 tumour expression have been described in different studies (*Carafa et al., 2019*; *Chen et al., 2014*; *Firestein et al., 2008*), which points to a discrepancy between SIRT1 level and activity. Second, SIRT1 deacetylates the VDR to enhance its activity in kidney and bone cells (*Sabir et al., 2017*) and, consequently, decreased SIRT1 activity may drive VD insensitivity. Third, promoter-bound VDR increases *SIRT1* gene expression in kidney and liver cells (*Yuan et al., 2022*) and thus, VD deficiency may decrease SIRT1 levels leading to reduced activity. However, whether VD alters SIRT1 activity per se and/or SIRT1 protein expression in CRC remains unclear.

Here, we reveal that 1,25(OH)$_2D_3$ induces specific SIRT1 activity at multiple levels in human CRC cells. Notably, liganded VDR post-translationally increases SIRT1 activity through auto-deacetylation. Mechanistically, SIRT1 K610 is identified as a critical player for 1,25(OH)$_2D_3$ enhanced VDR-SIRT1 interaction, SIRT1 auto-deacetylation and activation. This occurs independently of increased SIRT1 RNA and protein expression and elevated cofactor ($NAD^+$) levels. Discrepant SIRT1 level and activity may explain the controversial role of SIRT1 as tumor suppressor or tumor promoter in CRC. Our results provide a new mechanistic insight into the association of VD deficiency with CRC and suggest potential therapeutic benefit for SIRT1 activators in SIRT1-positive CRC.

## Results

### 1,25(OH)$_2$D$_3$ increases SIRT1 level via the VDR in CRC cells

The question of whether VD governs SIRT1 activity in CRC was first approached in two well characterized CRC cell lines, HCT 116 and HT-29. Immunofluorescence analyses (**Figure 1A**) revealed that SIRT1 protein level doubled in response to 1,25(OH)$_2$D$_3$, a result confirmed by Western blotting of nuclear extracts (**Figure 1B**). The increased expression of SIRT1 protein by 1,25(OH)$_2$D$_3$ could be explained by a twofold upregulation of SIRT1 RNA revealed by RT-qPCR (**Figure 1C**). The induction of *CYP24A1* (**Figure 1D**), a major VDR target gene in many cell types served as control for 1,25(OH)$_2$D$_3$ activity.

VDR expression is the main determinant of cell responsiveness to 1,25(OH)$_2$D$_3$ and is downregulated in a proportion of patients with advanced CRC, implying that these patients would probably not benefit from the anticancer effects of 1,25(OH)$_2$D$_3$ (**Ferrer-Mayorga et al., 2017**; **Larriba and Muñoz, 2005**; **Pálmer et al., 2004**; **Peña et al., 2005**). To model CRCs patients with low VDR tumour level or with VD deficiency, we used an HCT 116-derived cell line stably depleted of VDR using shRNA (hereafter named ShVDR) (**Larriba et al., 2011**). Induction of *SIRT1* gene expression by 1,25(OH)$_2$D$_3$ was confirmed by RT-qPCR analysis in ShControl but not in ShVDR cells, although basal SIRT1 levels were similar (**Figure 1E**). At the protein level, confocal imaging showed reduced SIRT1 protein in ShVDR as compared to ShControl cells (**Figure 1F**), suggesting that VDR contributes to maintaining basal SIRT1 protein expression. Western blot analyses on nuclear extracts confirmed that ShControl cells responded to 1,25(OH)$_2$D$_3$ with increased SIRT1 protein levels in contrast to the unresponsiveness of ShVDR cells (**Figure 1G**). Moreover, the upregulating effect of 1,25(OH)$_2$D$_3$ was specific for SIRT1, since the expression of the other nuclear sirtuin, SIRT7, was unaffected (**Figure 1H**). Together, these data indicated that 1,25(OH)$_2$D$_3$ specifically increased SIRT1 mRNA and protein levels in CRC cell lines via VDR. This is important because SIRT1 depletion and/or sirtuin activity decay might represent an important neglected mediator of VD deficiency and a target for intervention in CRC. Whether VD controls sirtuin activity independently of sirtuin level in CRC is a critical issue that deserved further exploration.

### 1,25(OH)$_2$D$_3$ induces SIRT1 deacetylase activity in CRC cells

Evaluation of global sirtuin activity in the nuclei of CRC cells revealed that treatment with 1,25(OH)$_2$D$_3$ increased NAD$^+$-dependent deacetylase activity 1.5-fold as compared to control untreated cells (**Figure 2A**). NAD$^+$ is a cofactor required for sirtuin activity. Metabolic control of the level of NAD$^+$ represents an important node for sirtuin regulation and could explain the induction of general sirtuin activity by 1,25(OH)$_2$D$_3$. Indeed, nuclear NAD$^+$ level doubled in response to 1,25(OH)$_2$D$_3$ in human CRC cells (**Figure 2B**). However, whether 1,25(OH)$_2$D$_3$ could specifically increase SIRT1 activity remained unknown.

Liganded VDR interacts with SIRT1 in kidney and bone cells (**Sabir et al., 2017**) suggesting that 1,25(OH)$_2$D$_3$ might alter SIRT1 expression or activity via post-transcriptional mechanisms. Since protein-protein interactions regulate stability and activity of proteins, the ability of 1,25(OH)$_2$D$_3$ to promote VDR-SIRT1 interaction in CRC cells was evaluated by co-immunoprecipitation assays. The results revealed that 1,25(OH)$_2$D$_3$ potently (4-fold) enhanced VDR/SIRT1 complex formation (**Figure 2C**). To test if this interaction alters SIRT1 activity, the acetylation status of two specific SIRT1 substrates (FOXO3a and H3K9) was evaluated as a read out of SIRT1 deacetylase activity in response to 1,25(OH)$_2$D$_3$. Acetyl-lysine immunoprecipitation followed by western blotting demonstrated efficient deacetylation of FOXO3a in response to 1,25(OH)$_2$D$_3$ (**Figure 2D**). Likewise, the level of acetylated histone H3K9 (Ace H3K9) dramatically decreased in response to 1,25(OH)$_2$D$_3$ (**Figure 2E**), indicating specific activation of SIRT1 by 1,25(OH)$_2$D$_3$. The effect of 1,25(OH)$_2$D$_3$ was mediated by VDR since ShVDR cells did not show Ace H3K9 deacetylation in response to 1,25(OH)$_2$D$_3$, whereas ShControl cells maintained the response (**Figure 2F**). Importantly, the specific SIRT1 activator SRT1720 reduced Ace H3K9 levels in both ShControl and ShVDR cells, thus mimicking 1,25(OH)$_2$D$_3$ action independently of VDR (**Figure 2F**).

The ability of 1,25(OH)$_2$D$_3$ to increase protein deacetylation mediated by SIRT1 could be driven either by elevated SIRT1 level or by increased SIRT1 activity. To distinguish between these two possibilities, we performed an in vitro deacetylase assay using equivalent amounts of immunoprecipitated SIRT1 from cells treated or not with 1,25(OH)$_2$D$_3$. SIRT1-specific deacetylase activity was

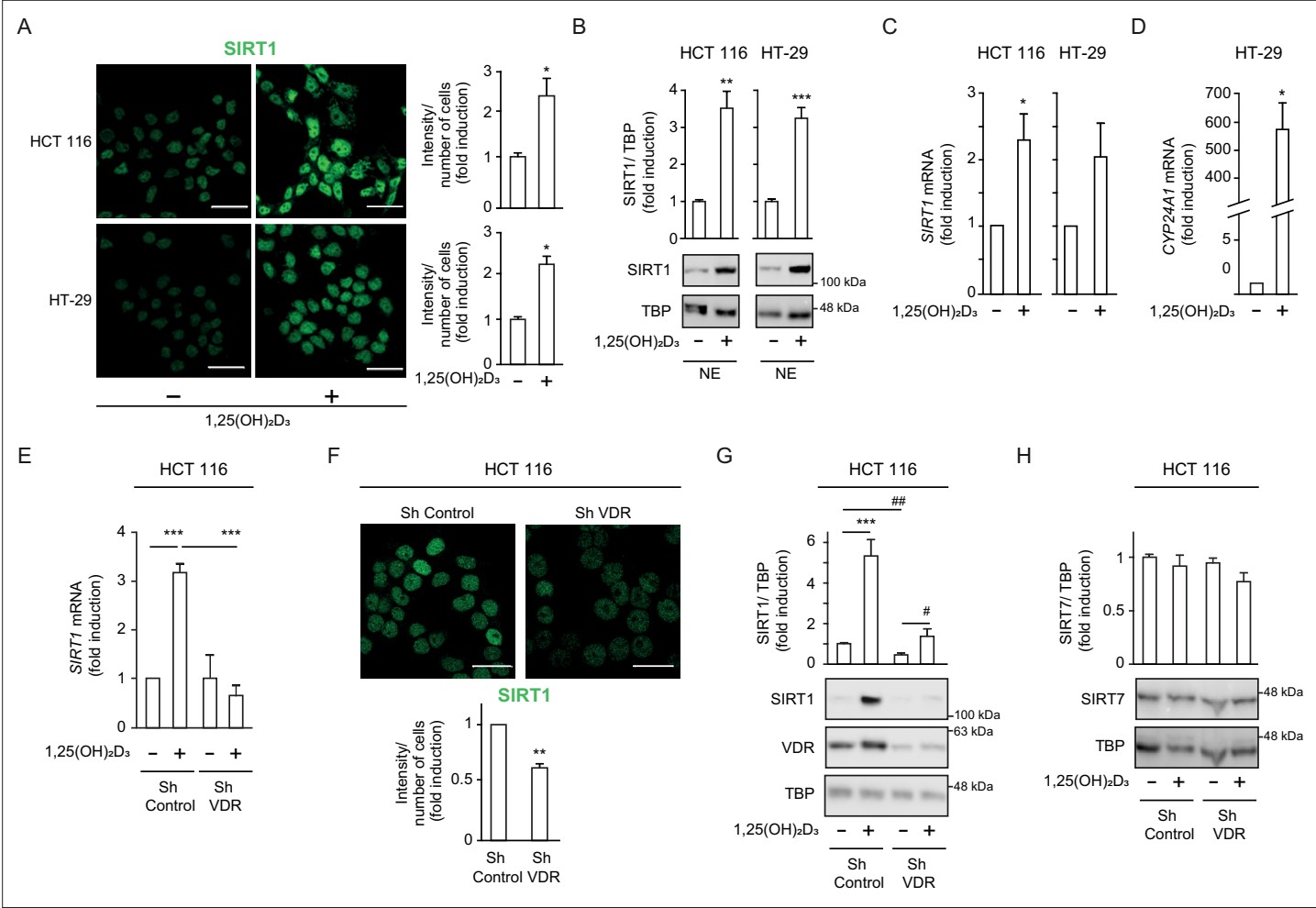

**Figure 1.** *1,25(OH)₂D₃ increases SIRT1 levels in CRC cells and VDR is required to ensure basal levels.* (**A**)-(**D**) HCT 116 or HT-29 CRC cells cultured under standard conditions and where indicated, treated with 1α,25-dihydroxyvitamin D₃ (1,25(OH)₂D₃), 100 nM, added 24 hr before harvesting. (**E**)-(**H**) ShVDR cells were derived from HCT 116 by stable and specific knock-down of vitamin D receptor (VDR) using specific shRNA and are compared to ShControl cells that contain normal VDR levels (*Larriba et al., 2011*). Cell extracts were fractionated in all cases. (**A**) Confocal imaging of SIRT1 (green) in CRC (HCT 116 and HT-29) cells and fluorescence intensity quantification of 3 independent experiments using ImageJ software. For each experiment, three different fields were evaluated per slide. Scale bars:25 µm. (**B**) Western-blot analysis of 1,25(OH)₂D₃ effects on levels of SIRT1 in nuclear extracts (NE) from indicated CRC cells. Representative blots and statistical analysis using TBP as loading controls. (**C**)-(**E**) RT-qPCR analysis of the effect of 1,25 (OH)₂D₃ in indicated colon cancer cells. Values normalized with endogenous control (18 S RNA) are referred as fold induction over cells without 1,25(OH)₂D₃. (**C**) Effect of 1,25 (OH)₂D₃ on SIRT1 gene expression. (**D**) validation of 1,25 (OH)₂D₃ activity on the canonical target gene cytochrome P450 family 24 subfamily A member 1, CYP24A1, in HT-29 colon cancer cells. (**E**) Requirement of vitamin D receptor (VDR) for 1,25 (OH)₂D₃ induction of SIRT1 gene expression. (**F**) Effect of VDR knock-down on SIRT1 protein content evaluated by confocal imaging of ShControl and ShVDR HCT 116 colon cancer cells to detect SIRT1 (green). Scale bars represent 25 µm. On the right, quantification of fluorescence intensity using ImageJ software; for each experiment, three different fields were evaluated per slide. (**G**) Western blot analysis of the effect of 1,25 (OH)₂D₃ on nuclear accumulation of SIRT1 in cells depleted (ShVDR) or not (ShControl) of VDR. (**H**) Specificity of 1,25 (OH)₂D₃ effects on SIRT1 by western blot analysis of the alternative nuclear sirtuin, SIRT7. Statistical analysis of three independent experiments in each panel was performed by Student t-test (**A**) to (**D**) and (**H**) or by One-Way ANOVA (**E**)-(**F**). For (**G**), statistical analysis of the four groups by ANOVA is represented by * and comparison of the two indicated groups by Students t test, by #. In all panels, values represent mean ± SEM of triplicates corresponding to biological replicates; * or # p<0.05; ** p<0.01; *** p<0.001. Raw data are available in *Figure 1—source data 1*.

The online version of this article includes the following source data for figure 1:

**Source data 1.** Presents results summarized in *Figure 1* (triplicates).

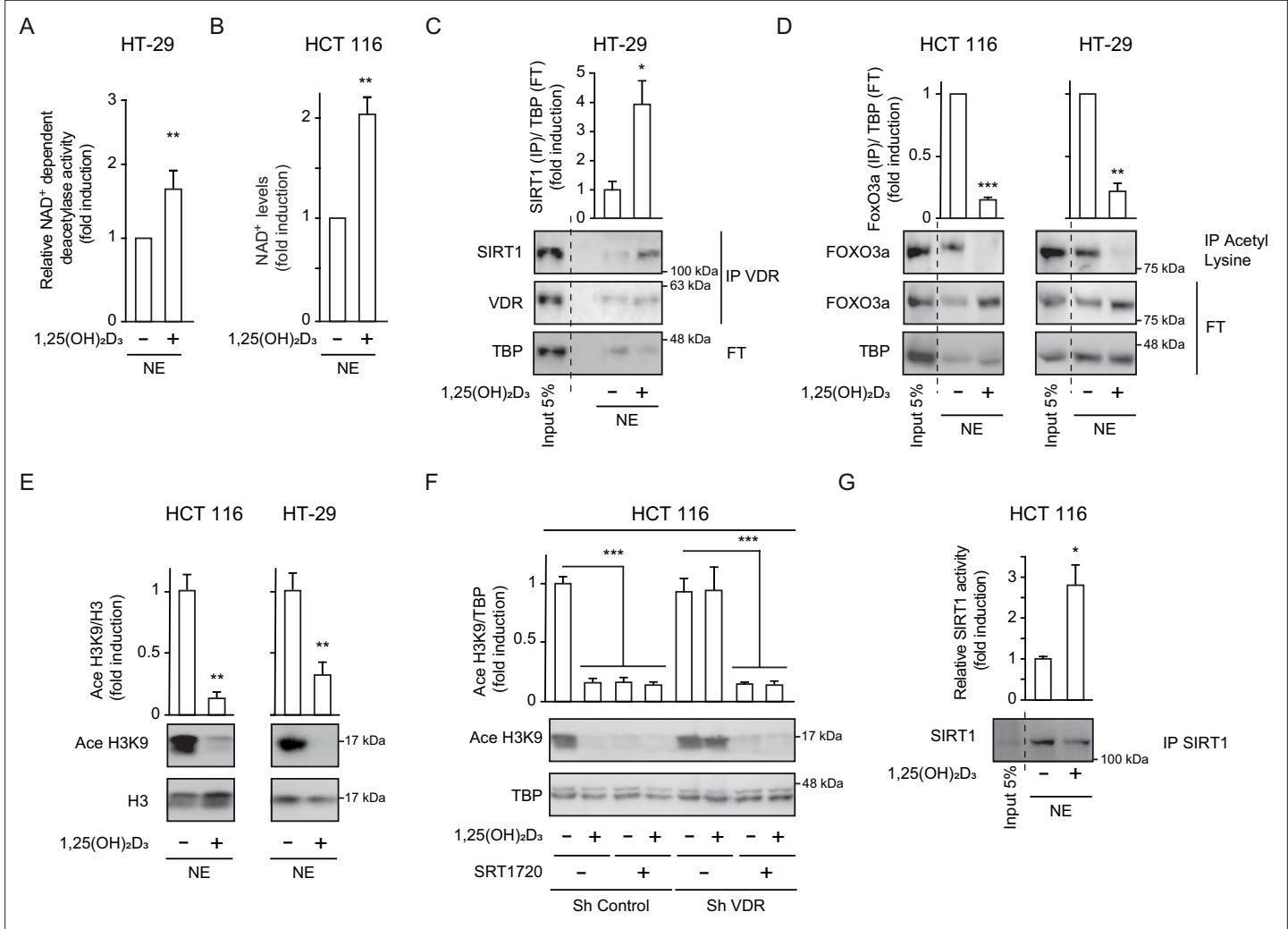

**Figure 2.** *1,25(OH)₂D₃ induces specific SIRT1 deacetylase activity independently of levels, in CRC cells.* Cells were cultured as in **Figure 1**. (**A**) and (**G**) NAD⁺-dependent deacetylase activity, (**B**) Evaluation of nuclear NAD⁺ levels. (**C**) to (**F**) Western blot analysis using TBP or H3 as loading controls. Representative western blots and statistical analysis of 3 independent experiments. (**A**) NAD⁺-dependent deacetylase activity measured on nuclear extracts of HT-29 cells. Relative luciferase units (RLU) were calculated as fold induction relative to the corresponding control. (**B**) Evaluation of nuclear NAD⁺ levels in HT-29 CRC cells in response to 1,25(OH)₂D₃, with data expressed as fold induction over the control, untreated cells. (**C**) 1,25(OH)₂D₃ effects on the interaction between vitamin D receptor (VDR) and SIRT1. Input (5%) and flow through (FT) are shown (**D**) 1,25(OH)₂D₃ effects on the acetylation status of the SIRT1 substrate FoxO3a in nuclear extracts immunoprecipitated with anti-acetyl lysine antibodies. Input (5%) and flow through (FT) are shown. (**E**) 1,25(OH)₂D₃ effects on the nuclear acetylation of the SIRT1 substrate H3K9 (Ace H3K9). (**F**) Requirement of VDR for deacetylation of Ace H3K9 in response to 1,25(OH)₂D₃ (**G**) Quantification of NAD⁺-dependent deacetylase activity on SIRT1 immunoprecipitates from nuclear extracts (NE) of HCT 116 CRC cells treated or not with 1,25(OH)₂D₃. Values represent mean ± SEM of n=3 independent experiments. Statistical analysis by Student t test, n≥3 biological replicates and values represent mean ± SEM; *p<0.05; **p<0.01; *** p<0.001. Raw data are available in **Figure 2—source data 1**.

The online version of this article includes the following source data for figure 2:

**Source data 1.** Contains all original data summarized in this figure.

enhanced threefold in cells treated with 1,25(OH)₂D₃ despite similar levels of immunoprecipitated SIRT1 (**Figure 2G**).

Together, these results indicated that 1,25(OH)₂D₃ induces nuclear SIRT1 activity. This implies that VD deficiency and/or loss of tumour VDR expression (both observed in a proportion of CRC patients) (**Evans et al., 1998**; **Ferrer-Mayorga et al., 2019**; **Giardina et al., 2015**; **Larriba et al., 2013**; **Larriba et al., 2009**; **Pálmer et al., 2004**; **Peña et al., 2005**) may negatively impact SIRT1 deacetylase activity and prompted us to examine samples of CRC patients.

## CRC biopsies exhibit discrepancies between SIRT1 protein levels and deacetylase activity

Changes in SIRT1 RNA and protein expression and enzymatic activity were analysed in samples of CRC patients. First, we compared available data (TCGA-COAD database) from 160 pairs of human colon adenocarcinoma samples matched with their normal adjacent tissue. TNM plot analysis revealed that both *VDR* and *SIRT1* gene expression decreased from normal to tumour colonic tissue in a highly significant fashion (p<0.001) (*Figure 3A–B*).

Immunohistochemical analysis of human colon tumor tissue microarrays (TMAs) from the Biobank of Hospital Clínico San Carlos (HCSC, Madrid) revealed a significant (p=0.017) positive association between the levels of VDR and SIRT1 proteins (*Figure 3C*). An additional TMA containing healthy colon samples was used to compare tumor and nontumor tissues. The level of VDR protein was usually lower in tumour samples (*Figure 3D*), while SIRT1 protein expression showed high variability between tumors and high cellular intratumor heterogeneity (*Figure 3D–E*). Globally, TMA data suggested that SIRT1 protein levels in tumor samples (*Figure 3E*), are subjected to high variability, which hampers statistical significance even though for SIRT1 gene expression, differences were statistically significant as shown in *Figure 3B*.

Global acetylation also showed high cellular intratumor heterogeneity and variability between samples without statistically significant differences between tumor and nontumor samples (data not shown), showing that is not a valid, specific read out of SIRT1 activity. As an alternative approach, the levels of SIRT1 protein and its substrate Ace H3K9 were compared first in intestinal cell lines and then in human tumours. Surprisingly, SIRT1 protein level increased in CRC cells as compared to normal HIEC6 cells (*Figure 3F*). However, increased SIRT1 levels coincided with elevated Ace H3K9 (*Figure 3F*), indicating SIRT1 inactivation in CRC cells. The results in cell lines encouraged us to evaluate the potential discordance of SIRT1 and Ace H3K9 levels in fresh-frozen samples from paired tumor and nontumor colonic tissue of patients from Hospital Universitario Fundación Alcorcón (HUFA). All tissues were evaluated to have at least 85% of tumor cells by immunohistochemistry (IHC). Western blotting allowed simultaneous detection of SIRT1 and Ace H3K9. Notably, whereas the level of SIRT1 protein variably increased or decreased from nontumor to tumor tissues in different patients, acetylation of its specific substrate H3K9 consistently increased in tumor samples (*Figure 3G*), reflecting SIRT1 inactivation as found in carcinoma cell lines (*Figure 3F*). In vitro deacetylation assays performed on SIRT1 immunoprecipitated from SIRT1-positive patient samples confirmed a common decrease of SIRT1 activity in colon tumor tissue (*Figure 3H*). Western blotting illustrated the variability of SIRT1 level among tumors and its correlation with VDR level (*Figure 3I*), while Ace H3K9 levels are generally high, but variable. Importantly, in vitro deacetylation assays on SIRT1 immunoprecipitated from the same samples (*Figure 3J*) confirmed Ace H3K9 as a *bona fide* read out of SIRT1 inactivation. The discrepancy between the levels of SIRT1 protein expression and activity in tumor samples may solve the controversy previously reported regarding the use of SIRT1 as a tumor biomarker and poses the question of how 1,25(OH)$_2$D$_3$ specifically enhances SIRT1 deacetylase activity.

## 1,25(OH)$_2$D$_3$ induction of SIRT1 activity is mediated through auto deacetylation

Since SIRT1 catalyses its auto deacetylation as a mechanism to increase its deacetylase activity towards other substrates (*Fang et al., 2017*) and 1,25(OH)$_2$D$_3$ facilitates VDR-SIRT1 interaction, we explored the possibility that liganded VDR increases SIRT1 activity by facilitating SIRT1 auto deacetylation. To evaluate SIRT1 acetylation status upon exposure to 1,25(OH)$_2$D$_3$, nuclear extracts from CRC cells challenged or not with 1,25(OH)$_2$D$_3$ were immunoprecipitated using anti-acetyl-lysine antibodies and subjected to Western blotting. 1,25(OH)$_2$D$_3$ greatly reduced the nuclear pool of acetylated SIRT1 (*Figure 4A*). Remarkably, the specific SIRT1 activator SRT1720 also drove SIRT1 deacetylation in HCT 116 CRC cells (*Figure 4B*). Consistently, ShVDR cells did not reduce acetylated SIRT1 in response to 1,25(OH)$_2$D$_3$ but retain their response to SRT1720 (*Figure 4C*). These results indicated that 1,25(OH)$_2$D$_3$ activation of SIRT1 is mediated through SIRT1 deacetylation.

Next, we sought to get further insight into the mechanism of regulation of SIRT1 activity by 1,25(OH)$_2$D$_3$. As the acetyltransferase EP300 acetylates and inactivates SIRT2 (*Han et al., 2008*) and is critical for CRC cell signalling (*Chocarro-Calvo et al., 2013*; *Gutiérrez-Salmerón et al., 2020*), we used the ASEB engine (*Wang et al., 2012*) to search for SIRT1 residues potentially acetylated by

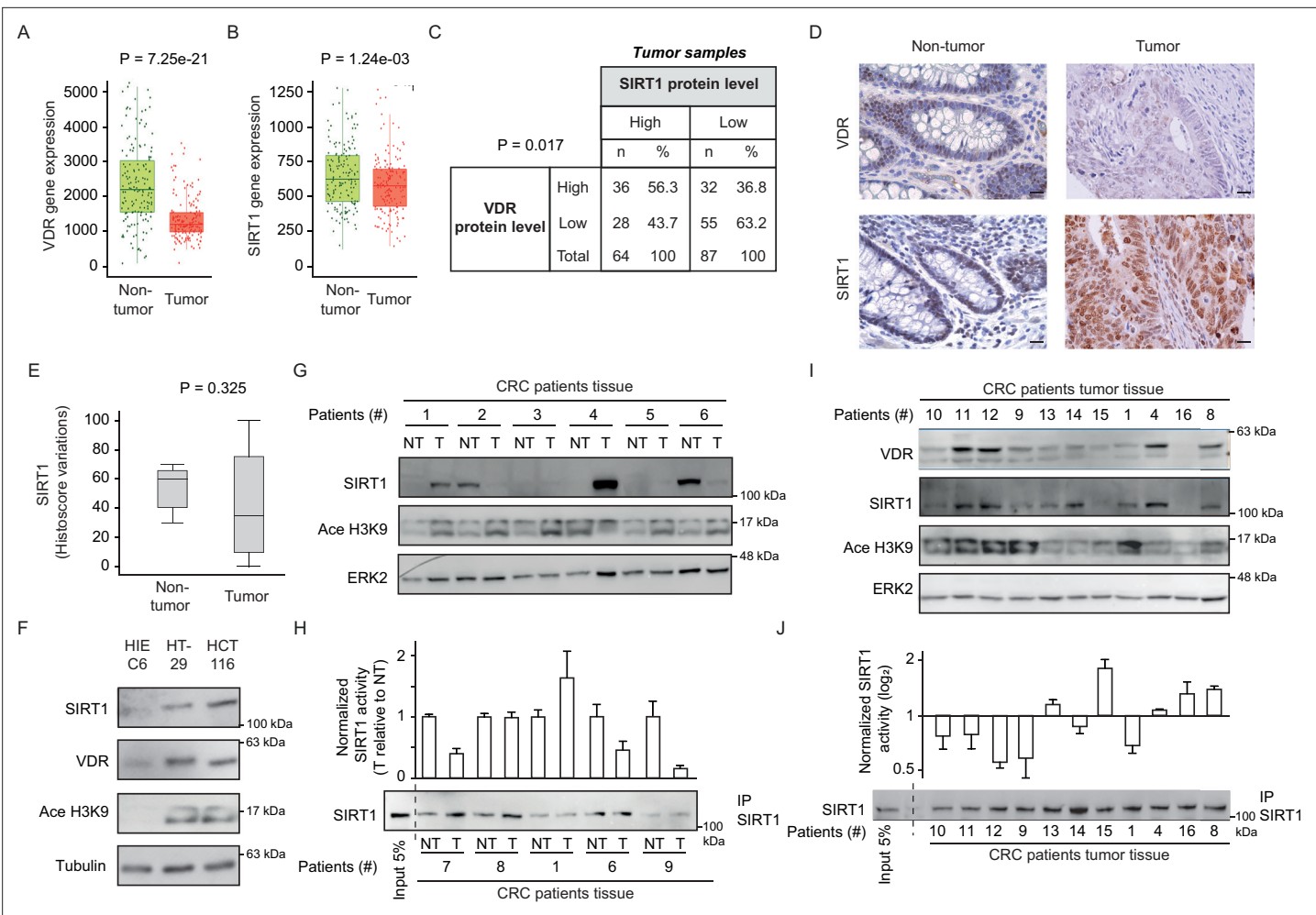

**Figure 3.** CRC biopsies exhibit discrepancies between SIRT1 protein levels and deacetylase activity. (**A**)-(**B**) TNM plot analysis of VDR (**A**) or SIRT1 (**B**) gene expression in 160 paired samples of non-cancer adjacent tissue (normal) versus adenocarcinoma tissue (tumour). Data was collected from TCGA-COAD database. SIRT1 or VDR expression is shown as transcripts per million (TPM) in log2. Tumour and control samples were compared with Mann-Whitney U test. The statistical significance cut off was set at p<0.01. (**C**) Association analysis between VDR and SIRT1 protein levels in human colon cancer samples, stage 2 and 4. Cut-off points to separate proteins between high- or low- expression levels were for SIRT1 expression median of Histoscore, and negative *versus* positive staining for VDR expression. Statistical analysis was performed with Chi-square test. (**D**) Representative micrographs at ×40 from immunostaining for vitamin D receptor (VDR) or SIRT1 on human samples of healthy colon or CRC patients. Scale bars: 20 μm. Immunostainings were revealed with DAB (diaminebenzidine) and thus, positiveness is highlighted by light or dark brown according to their low or high protein expression. Counterstaining with hematoxilin stains nuclei in dark blue and cytoplasm in light blue. (**E**) Profile for SIRT1 content from healthy to colon cancer human samples. The profile was obtained from Histoscore variations. Tumour and control samples were compared with Mann-Whitney U test. The statistical significance cut off was set at p<0.01. (**F**) Western-blot analysis of the levels of SIRT1 and its substrate Ace H3K9 in total lysates from intestinal healthy (HIEC6) or CRC cells. Representative western blot with ERK as loading control. (**G**) Western-blot analysis from human patient samples, with patient number indicated at the top. CRC samples (**T**) and adjacent nontumor tissue (NT) were probed for SIRT1 and its substrate Ace H3K9 (as in F). Frozen samples were obtained from HUFA patients. (**H**) NAD[+]-dependent deacetylase activity on immunoprecipitated SIRT1 from patient lysates. Values for T and NT samples were corrected according to immunoprecipitated SIRT1 levels shown on the western underneath the graph. Values for T samples were referred to values of NT and data (duplicates) are expressed as fold induction. Patient number is indicated at the bottom. (**I**) Western-blot analysis for SIRT1 and its substrate Ace H3K9 levels in total lysates from alternative human CRC samples (as in G). Patient number is indicated at the top. (**J**) NAD[+]-dependent deacetylase activity on immunoprecipitated SIRT1 from patient lysates presented in (**I**). Levels of immunoprecipitated SIRT1 are presented in the bottom panel. SIRT1 deacetylase activity in patients is presented as fold induction, referred to the first patient sample (patient #10). Patient number is indicated at the bottom. Raw data are available in *Figure 3—source data 1*.

The online version of this article includes the following source data for figure 3:

**Source data 1.** Contains all original data summarized in this figure.

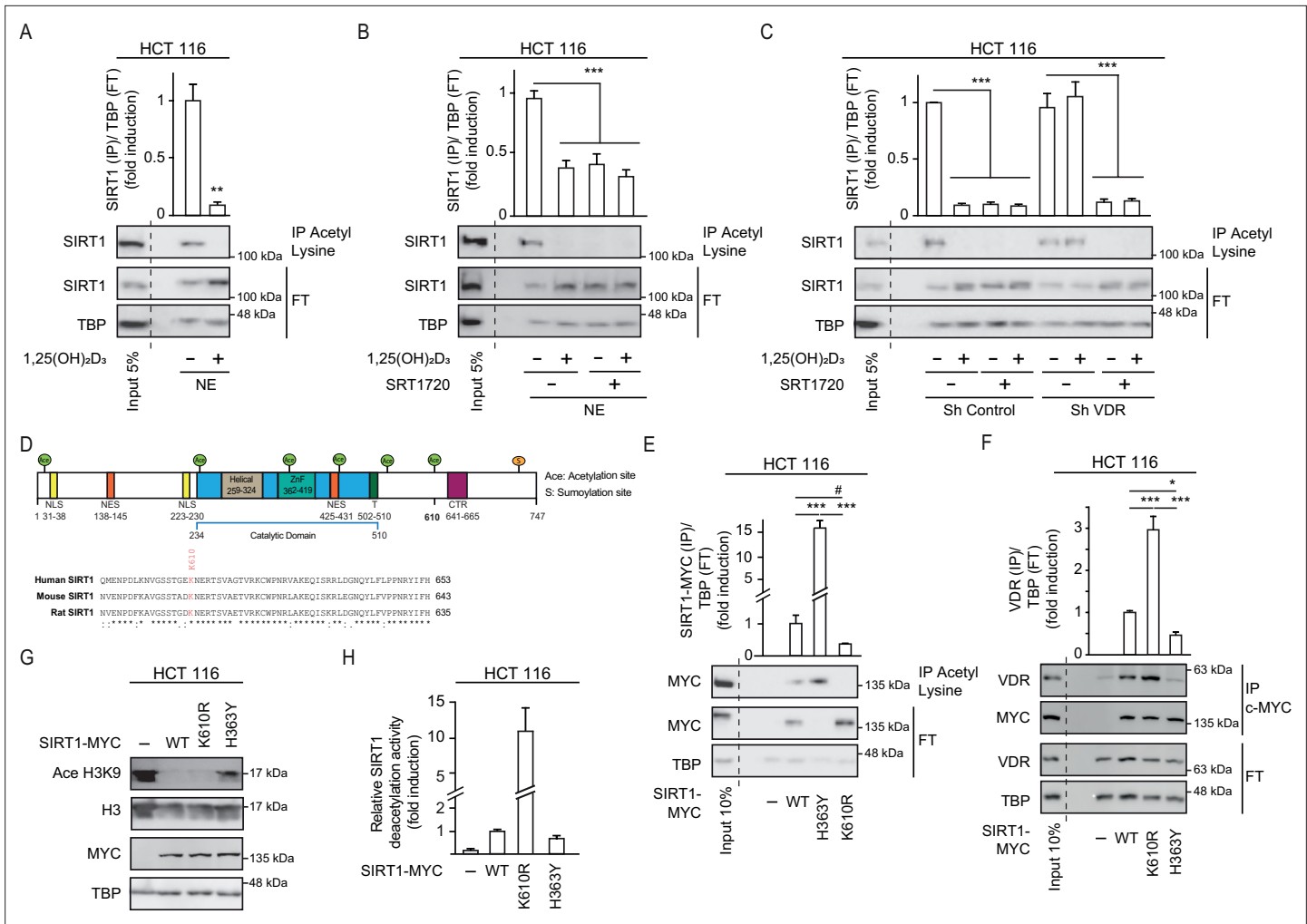

**Figure 4.** 1,25(OH)₂D₃ induces SIRT1 activity through auto deacetylation. HCT 116 CRC cells were cultured under standard conditions in DMEM containing LiCl (40 mM) to mimic Wnt signalling before addition of 1,25(OH)₂D₃, 100 nM or SRT1720 (20 μM) for the last 24 hr; after fractionation nuclear extracts (NE) were used. (**A**)-(**C**) and (**E**)-(**F**) Immunoprecipitation (IP) using anti-acetyl lysine antibody from nuclear extracts (NE) of indicated cells and western-blot analysis to evaluate changes in acetylation of SIRT1; TBP was used in the Flow Through (FT) as loading control. Representative western-blots and statistical analysis are shown in all panels. (**A**) Effect of 1,25(OH)₂D₃ on acetylation of SIRT1. (**B**) Effect of SRT1720 on acetylation of SIRT1. (**C**) Requirement of VDR for 1,25(OH)2D₃ or SRT1720 effects on acetylation of SIRT1 in Sh Control and ShVDR HCT 116 cells. (**D**) SIRT1 scheme with functional domains and putative acetylation sites (top) and sequence alignment. Conserved putative acetylation targets for activation/inactivation of SIRT1 shown at the bottom. (**E**) Acetylation status of exogenous SIRT1 wild type (WT) or mutants (H363Y or K610R). HCT 116 were transiently transfected with pcDNA expression vectors: empty (-), myc-tagged SIRT1 wild type (WT) or mutants: H363Y inactive or K610R, 48 hr before harvesting. Immunoprecipitation using anti-acetyl-lysine antibodies and western blot detection using myc antibodies. Representative western-blot and statistical analysis; MYC and TBP in the flow through (FT) serve as loading controls. (**F**) Western-blot analysis of the interaction of SIRT1 wild type and mutants, with vitamin D receptor (VDR). Expression and analysis as in (**E**). (**G**) Western-blot analysis of the effect of SIRT1 mutant expression on Ace H3K9. (**H**) In vitro NAD⁺-dependent deacetylase activity assays on equivalent amounts of exogenous SIRT1 wild type (WT), K610R and inactive H363Y mutants immunoprecipitated using anti-myc antibodies. Statistical analysis in all panels except (**D**) and (**G**) by One Way ANOVA of n>3 independent experiments. Values represent mean ± SEM; *p<0.05; **p<0.01; *** p<0.001. Raw data are available in *Figure 4—source data 1*.

The online version of this article includes the following source data for figure 4:

**Source data 1.** Contains all original data summarized in this figure.

EP300. This approach identified K610, a lysine conserved from mouse to human, as a potential EP300 target (*Figure 4D*). The role of K610 in the regulation of SIRT1 activity was assessed by generating a non-acetylable K610R mutant, which should be resistant to inactivation through acetylation. We compared the enzymatic activity of K610R SIRT1 with that of wild type (WT) and inactive H363Y SIRT1 (*Vaziri et al., 2001*). Nuclear extracts from cells transfected with MYC-tagged SIRT1 WT or mutants

were immunoprecipitated using anti-acetyl-lysine antibodies. Western blotting using an anti-MYC antibody revealed that exogenous WT SIRT1 was acetylated (*Figure 4E*), as it was endogenous SIRT1 (*Figure 4A*). Of note, catalytically inactive H363Y SIRT1 was highly acetylated, in striking contrast with the lack of acetylation of the K610R SIRT1 mutant (*Figure 4E*), This highlights the importance of K610 for the regulation of SIRT1 activity.

Since interaction with VDR was important for SIRT1 deacetylation activity, the ability of each mutant to interact with VDR was compared in immunoprecipitation assays. The results revealed that the K610R SIRT1 mutation facilitated the interaction with VDR, whereas the inactive H363Y SIRT1 showed low interaction with VDR (*Figure 4F*). This suggested that acetylation of SIRT1 might interfere with or destabilize its interaction with VDR, and that the ability of 1,25(OH)$_2$D$_3$ to favour VDR-SIRT1 interaction may allow SIRT1 auto deacetylation, and thus enhance its activity. The activity of mutant SIRT1 enzymes was first inferred from the level of Ace H3K9, which was lower in cells expressing K610R SIRT1 (*Figure 4G*). In vitro NAD$^+$-dependent deacetylase assays were next used to compare the activity of WT and mutant SIRT1. Using equivalent amounts of immunoprecipitated MYC-tagged enzymes, we found a 10-fold higher deacetylase activity for the K610R mutant than for WT SIRT1, whereas as expected the H363Y inactive mutant exhibited very low activity as expected (*Figure 4H*).

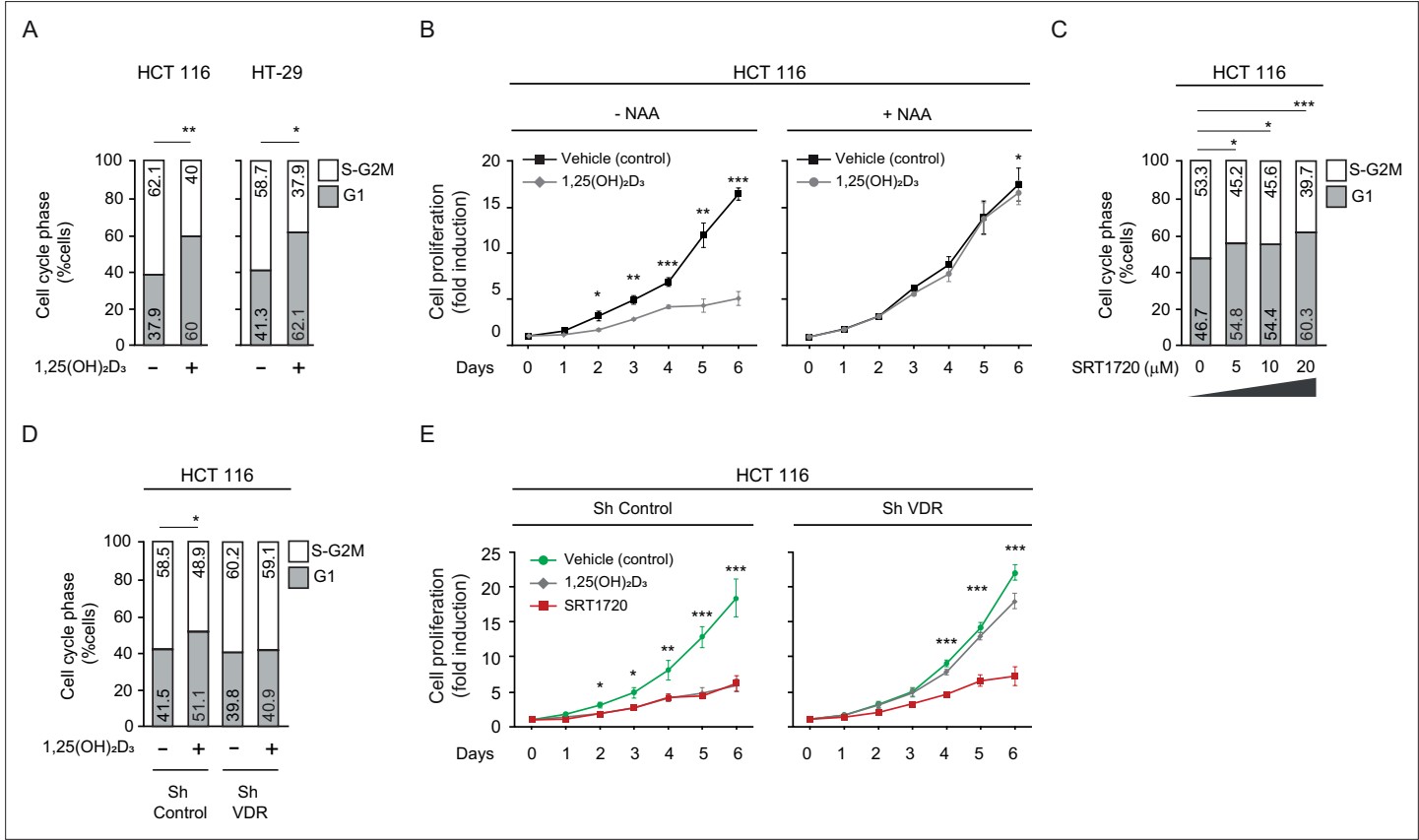

**Figure 5.** SIRT1 activation rescues antiproliferative effects of 1,25(OH)$_2$D$_3$ in unresponsive CRC cells. Cells were cultured as in previous figures and treatments with NAA (300 uM), SRT1720 (20 µM) and 1,25 (OH)$_2$D$_3$ (100 nM) also were for the last 24 hr. (**A**) Flow cytometry analysis of cell cycle from indicated cells treated (+) or not (-) with 1,25(OH)$_2$D$_3$. Numbers correspond to the percentage of cells in the indicated phases expressed as mean ± SEM. (**B**) Effect of NAA on the 1,25(OH)$_2$D$_3$-driven blockades of HCT 116 CRC cell proliferation. (**C**) Effects of SRT1720 on cell cycle of HCT 116 CRC cells by flow cytometry analysis. Numbers correspond to the percentage of cells in the indicated phases expressed as mean ± SEM. (**D**) Flow cytometry analysis of cell cycle response to 1,25 (OH)$_2$D$_3$ in ShControl and ShVDR HCT 116 cells. Numbers correspond to the percentage of cells in the indicated phases expressed as mean ± SEM. (**E**) Proliferation curves of ShControl or ShVDR HCT 116 colon cancer cells in response to 1,25 (OH)$_2$D$_3$ (100 nM) (grey), compared to control with vehicle (green) or SRT1720 (red). Statistical analysis by One-Way ANOVA of three independent experiments; values represent mean ± SEM of triplicates; * p<0.05; ** p<0.01; *** p<0.001.

## SIRT1 activation mimics the antiproliferative effects of 1,25(OH)$_2$D$_3$ in CRC cells unresponsive to VD

Collectively, our data indicated that the acetylation status of SIRT1 K610 is critically controlled and governs SIRT1 activity in the nuclei of CRC cells. By promoting VDR-SIRT1 interaction, 1,25(OH)$_2$D$_3$ facilitates SIRT1 auto deacetylation and enhances SIRT1 activity. These results suggest that SIRT1 is an effector of 1,25(OH)$_2$D$_3$ and prompted us to ask whether the pharmacological activation of SIRT1 would rescue CRC cells that are unresponsive to 1,25(OH)$_2$D$_3$.

The anticancer effects of 1,25(OH)$_2$D$_3$ are exerted at least partially through the inhibition of tumor cell proliferation (*Carlberg and Muñoz, 2022a*; *Muñoz and Grant, 2022*). 1,25(OH)$_2$D$_3$ strongly reduced the percentage of CRC cells in S-G$_2$M cell-cycle phases (*Figure 5A*) and consequently, it lengthened the cell cycle and reduced CRC cell proliferation. We aimed to analyse the effect of the modulation of SIRT1 activity on the proliferation of CRC cells and on the antiproliferative action of 1,25(OH)2D3. Inhibition of general sirtuin activity with Nicotinamide (NAA) abolished the antiproliferative effects of 1,25(OH)$_2$D$_3$ on CRC cells (*Figure 5B*). Moreover, activation of SIRT1 by the small molecule SRT1720 mimicked the response to 1,25(OH)$_2$D$_3$ and efficiently reduced the percentage of cells in S-G$_2$M cell cycle phases (*Figure 5C*). These data indicate that induction of SIRT1 activity mediates 1,25(OH)$_2$D$_3$-driven cell cycle lengthening in CRC cells.

As expected, in contrast with ShControl cells, ShVDR CRC cells did not extend the cell cycle in response to 1,25(OH)$_2$D$_3$ (*Figure 5D*). Accordingly, an early and strong reduction in the proliferation rate in response to 1,25(OH)$_2$D$_3$ was noted in ShControl cells but it was almost absent in ShVDR cells (*Figure 5E*). Specific SIRT1 activation with SRT1720 reduced the proliferation of ShControl cells at a comparable magnitude to 1,25(OH)$_2$D$_3$ and, importantly, also reduced proliferation of 1,25(OH)$_2$D$_3$-unresponsive ShVDR cells (*Figure 5E*).

Collectively, these results show that SIRT1 acts downstream of 1,25(OH)$_2$D$_3$-activated VDR in the inhibition of CRC cell proliferation. They also suggest that activation of SIRT1 in CRC patients with vitamin D deficiency or with VD-unresponsive tumors (conditions frequently observed in advanced CRC), could be a therapeutic strategy to effectively antagonize cell proliferation.

## Discussion

Here, we reveal that 1,25(OH)$_2$D$_3$ activates the epigenetic regulator SIRT1 by acting at multiple levels. Ligand-activated VDR post-translationally upregulates SIRT1 activity by facilitating its auto deacetylation. Mechanistically, SIRT1 K610 is identified as essential to control VDR-SIRT1 interaction and SIRT1 acetylation status and activity. Notably, both SRT1720 and 1,25(OH)$_2$D$_3$ activate SIRT1 through auto deacetylation. Modulation of SIRT1 activity by 1,25(OH)$_2$D$_3$ through deacetylation occurs in addition to increased mRNA and protein levels as well as to an elevation of cofactor (NAD$^+$) availability. Thus, 1,25(OH)$_2$D$_3$ activates SIRT1 acting at multiple levels, from gene expression to activity, and these findings broaden our understanding of how VD deficiency is associated with CRC.

We also show that SIRT1 deacetylase activity mediates the anti-proliferative action of 1,25(OH)$_2$D$_3$ in CRC, which is in line with the recent proposal that SIRT1 activity opposes several cancer cell hallmarks (*Yousafzai et al., 2021*). In cases of VD deficiency and/or unresponsiveness such as advanced colorectal tumor lacking VDR expression due to transcriptional repression by SNAIL1/2 (*Pálmer et al., 2004*; *Larriba et al., 2009*), the activation of SIRT1 by compounds other than 1,25(OH)$_2$D$_3$ may be especially relevant. Clinical trials that explore SIRT1 activators in CRC are rare (https://clinicaltrials.gov). Some trials are designed to study doses and pharmacokinetics of resveratrol, a known sirtuin activator that has many other targets (*Patel et al., 2011*). Currently there are not known clinical trials using SRT1720 or alternative SIRT1 specific activators.

Consideration of SIRT1 protein level as a prognostic marker for cancer has received much attention, but conflicting results reporting both, tumor SIRT1 increases and decreases raised controversy (*Ren et al., 2017*; *Wu et al., 2017*). Accordingly, some studies propose SIRT1 either as tumour suppressor or as tumor promoter in CRC (*Carafa et al., 2019*; *Ren et al., 2017*). Importantly, SIRT1 protein level might not always reflect SIRT1 activity, and some tumors could possibly increase SIRT1 protein expression to counteract inactivation. Our results, which indicate the existence of discrepancies between SIRT1 levels and activity in human colorectal tumors, may help to resolve the controversy. Interestingly, alternative posttranslational SIRT1 modifications may have a similar outcome. For

example, proteolytic cleavage of SIRT1 induced by TNFα in osteoarthritis renders accumulation of a catalytically inactive 75 kDa resistant fragment (*Oppenheimer et al., 2012*) that could be detected by immunohistochemistry. Since inflammation (in general) and especially TNFαis an important mediator of CRC, understanding if this fragment is generated in CRC patients and whether its presence would also lead to decreased SIRT1 activity in samples strongly positive for SIRT1 remains as an interesting unexplored issue.

The ability of 1,25(OH)$_2$D$_3$ to activate SIRT1 may have further major implications for physiological and pathological processes other than CRC since SIRT1 target proteins control several processes such as mitochondria physiology, apoptosis, and inflammation (*Strycharz et al., 2018*). Thus, 1,25(OH)$_2$D$_3$ may not only regulate gene expression via direct transcriptional regulation of VDR-bound targets, but it can secondarily impact a wide range of events and processes via the control of SIRT1 activity. This may explain how 1,25(OH)$_2$D$_3$ regulates the activity of FOXO3a, an important modulator of cell proliferation and the immune response that is targeted by SIRT1 (*An et al., 2010*). SIRT1 also modulates the activity of the tumour suppressor p53, whose mutation is a relatively early event in a high proportion of CRCs (*Polidoro et al., 2013*; *Strycharz et al., 2018*). In this way, SIRT1 activation greatly broaden the known vitamin D regulatory action, albeit whether 1,25(OH)$_2$D$_3$ directs SIRT1 activity towards specific substrates remains to be elucidated. The accumulating evidence for SIRT1 involvement in chronic inflammation (*Mendes et al., 2017*) suggests a potential benefit of SIRT1 activation by 1,25(OH)$_2$D$_3$ in chronic inflammation-linked CRC and other diseases. Moreover, the mechanism revealed here for SIRT1 activation by 1,25(OH)$_2$D$_3$ may be relevant for diseases associated with VD deficiency beyond CRC, including autoimmune disorders and diabetes.

In conclusion, the multilevel activation of SIRT1 deacetylase by 1,25(OH)$_2$D$_3$ broadens the range of known 1,25(OH)$_2$D$_3$ targets and positions SIRT1 as an important mediator of the protective action of VD against CRC and potentially other neoplasias and non-tumoral diseases. These data support SIRT1 activators as useful agents in situations of dysfunction of the VD system.

## Materials and methods

**Key resources table**

| Reagent type (species) or resource | Designation | Source or reference | Identifiers | Additional information |
|---|---|---|---|---|
| Strain, (*Escherichia coli*) | *E. coli* 5α | New England Biolabs | NEB5α | |
| Cell line (*Homo-sapiens*) | HCT 116 | ATCC | Cat# CCL-247, RRID:CVCL_0291 | Male. Colorectal adenocarcinoma |
| Cell line (*Homo-sapiens*) | HT-29 | ATCC | Cat# HT B−38, RRID:CVCL_0320 | Female. Rectosigmoid adenocarcinoma |
| Cell line (*Homo-sapiens*) | HIEC6 | ATCC | Cat# CRL-3266 RRID:CVCL_6C21 | Epithelial. Intestinal. |
| Cell line (*Homo-sapiens*) | HCT 116 ShControl | In house; *Larriba et al., 2011* | | Male. Colorectal adenocarcinoma |
| Cell line (*Homo-sapiens*) | HCT 116 ShVDR | In house; *Larriba et al., 2011* | | Male. Colorectal adenocarcinoma |
| Antibody | anti-Acetylated-lysine Rabbit Polyclonal | Cell Signaling | Cat# 9441 | 1:1000 |
| Antibody | anti-Histone H3K9-Ace. Rabbit monoclonal | Cell Signaling | (C5B11) Cat # 9649 | 1:1000 |
| Antibody | anti-Histone H3 Rabbit Polyclonal | Cell Signaling | Cat # 9715 | 1:1000 |
| Antibody | anti-Vitamin D3 receptor (D2K6W) Rabbit Polyclonal | Cell Signaling | Cat #12550 | 1:1000 |

*Continued on next page*

*Continued*

| Reagent type (species) or resource | Designation | Source or reference | Identifiers | Additional information |
|---|---|---|---|---|
| Antibody | anti-FoxO3a (75D8) Rabbit Polyclonal | Cell Signaling | Cat# 2497 | 1:1000 |
| Antibody | anti-SIRT1 (D1D7) Rabbit Polyclonal | Cell Signaling | Cat# 9475 | 1:1000 |
| Antibody | anti-SIRT1. Rabbit Polyclonal | Santa Cruz Biotechnology | Cat#SC-15404 | 1:1000 |
| Antibody | anti-TFIID-TBP (N12). Rabbit Polyclonal | Santa Cruz Biotechnology | Cat# SC-204 | 1:1000 |
| Antibody | anti-SIRT7. Rabbit Polyclonal | Santa Cruz Biotechnology | Cat#SC365344 | 1:1000 |
| Antibody | anti-ERK (C14). Rabbit Polyclonal | Santa Cruz Biotechnology | Cat# SC-154 | 1:1000 |
| Antibody | anti-MYC (9E10). Mouse monoclonal | Santa Cruz Biotechnology | Cat# SC-40 | 1:1000 |
| Antibody | Anti-Rabbit- AlexaFluor488. Donkey Polyclonal. | Invitrogen | Cat# A21206 | 1:500 |
| Antibody | anti-TBP (51841). Mouse monoclonal | Invitrogen | Cat# MA514739 | 1:1000 |
| Antibody | Anti-Rabbit IgG (H+L) HRPO. Goat Polyclonal | BIO-RAD | Cat #170–6515 | 1:5000 |
| Antibody | Anti-Mouse IgG (H+L) HRPO. Goat Polyclonal | BIO-RAD | Cat #170–6516 | 1:5000 |
| Antibody | anti-Tubulin Clone B-5-1-2. Mouse monoclonal | Sigma-Aldrich | Cat# T5168 | 1:5000 |
| Recombinant DNA reagent | pcDNA3.1-SIRT1-Myc-His (plasmid) | Gift from Prof. Colin Goding. Oxford. UK | N/A | human |
| Recombinant DNA reagent | pcDNA3.1-SIRT1H363Y-Myc-His | Gift from Prof. Colin Goding. Oxford. UK | N/A | human |
| Recombinant DNA reagent | pcDNA3.1-SIRT1 K601R-Myc-His | This study | N/A | human |
| Sequence-based reagent | Human SIRT1 (F 5′–3′) | Sigma | K610R mutagenesis primers | GGTTCTAGTACTGGGGAGAGG AATGAAAGAACTTCAGTGG |
| Sequence-based reagent | Human SIRT1 (R 5′–3′) | Sigma | K610R mutagenesis primers | CCAGCCACTGAAGTTCTTTCA TTCCTCTCCCCAGTACTAG |
| Sequence-based reagent | Human CYCLIN D1 (CCND1). F 5′–3′: | Sigma | qPCR primers | AAGATCGTCGCCACCTGG |
| Sequence-based reagent | Human CYCLIN D1 (CCND1). R 5′–3′: | Sigma | qPCR primers | GGAAGACCTCCTCCTCGCAC |
| Sequence-based reagent | Human c-MYC F 5′–3′: | Sigma | qPCR primers | CTTCTCTCCGTCCTCGGATTCT |
| Sequence-based reagent | Human c-MYC R 5′–3′: | Sigma | qPCR primers | GAAGGTGATCCAGACTCTGACCTT |
| Sequence-based reagent | Human 18 s | Sigma | qPCR primers | AGTCCCTGCCCTTTGTACACA |
| Sequence-based reagent | Human 18 s | Sigma | qPCR primers | GCCTCACTAAACCATCCAATCG |
| Sequence-based reagent | CYP24A1 | Applied Biosystems | TaqMan probe | Hs00167999_m1 |

*Continued on next page*

*Continued*

| Reagent type (species) or resource | Designation | Source or reference | Identifiers | Additional information |
|---|---|---|---|---|
| Commercial assay or kit | EnVision +Dual Link System-HRP (DAB+) | Dako (Agilent) | Cat#K4065 | |
| Commercial assay or kit | SIRT-Glo Assay kit | Promega | G6450 | |
| Commercial assay or kit | NAD/NADH-Glo Assay Kit | Promega | G9071 | |
| Commercial assay or kit | QuickChange Site-Directed Mutagenesis Kit | Stratagene | Cat #200523 | |
| Peptide, recombinant protein | DYNAbeads Protein A | Invitrogen | Ref 10002D | |
| Peptide, recombinant protein | DYNAbeads Protein G | Invitrogen | Ref 10004D | |
| Chemical compound, drug | JetPEI PolyPlus reagent | Genycell Biotech | Cat# 101–10 N | |
| Chemical compound, drug | JetPRIME PolyPlus reagent | Genycell Biotech | Cat # 114–01 | |
| Chemical compound, drug | DMEM media | Lonza | Cat# 12–604 F | |
| Chemical compound, drug | Bovine Fetal Serum | Sigma | Cat# F7524 | |
| Chemical compound, drug | 7-AAD | Santa Cruz Biotechnology | SC-221210 | |
| Chemical compound, drug | 3-(4,5-dimetiltiazol-2-il)–2,5-difeniltetrazol (MTT) | Sigma-Aldrich | Cat# M5655 | |
| Chemical compound, drug | BSA | Sigma-Aldrich | A7906 | |
| Chemical compound, drug | Nicotinamide (NAA) | Sigma-Aldrich | N3376 | |
| Chemical compound, drug | SRT1720 | Selleckchem | S1129 | |
| Chemical compound, drug | TRIzol reagent | Invitrogen | Cat#15596026 | |
| Chemical compound, drug | Protease inhibitor Cocktail | Roche | Cat#04693132001 | |
| Chemical compound, drug | $1\alpha,25$-DihydroxyVD$_3$ | Sigma-Aldrich | 17936 | |

*Continued on next page*

*Continued*

| Reagent type (species) or resource | Designation | Source or reference | Identifiers | Additional information |
|---|---|---|---|---|
| Chemical compound, drug | Lithium Chloride | Sigma-Aldrich | Cat# L9650 | |
| Software, algorithm | LAS AF software | Leica | SP5 | |
| Software, algorithm | 3730xl Analyzer | Applied Biosystem | ABI 3730XL | |
| Software, algorithm | GraphPad Prism software | GraphPad Software | https://www.graphpad.com | |
| Software, algorithm | ImageLab | Bio-Rad | ChemiDoc XRS +System | |
| Software, algorithm | CXP software | Becton-Dickinson | FACSCalibur | |
| Software, algorithm | ImageJ software | https://imagej.nih.gov/ij/download.html | N/A | |
| Software, algorithm | SPSS Statistics version 26 | IBM | N/A | |

## Contact for reagent and resource sharing

Further information and requests for reagents may be directed to, and will be fulfilled by Lead Contact: Dr. Custodia García-Jiménez: custodia.garcia@urjc.es.

## Experimental model and subject details

### Human samples

Patient samples were derived from surgical removal at Fundación Jimenez Diaz University Hospital, General and Digestive Tract Surgery Department: 95 patient samples diagnosed with stage II CRC and 56 with stage IV from primary site. The Institutional Review Board (IRB) of the Fundación Jimenez Diaz Hospital, reviewed and approved the study, granting approval on December 9, 2014, act number 17/14. Samples from seven healthy individuals were obtained from Hospital Clínico San Carlos (HCSC). IRB-HCSC act number 21/498-E granting approval on June 25th, 2021. Paired fresh-frozen samples from tumor and nontumor adjacent tissue were collected from 16 patients at Hospital Universitario Fundación Alcorcón (HUFA). IRB-HUFA, act number 17/68 granting approval on June 8th, 2017. Colon cancer tissues were collected using protocols approved by the corresponding Ethics Committees and following the legislation. All patients gave written informed consent for the use of their biological samples for research purposes. Fundamental ethical principles and rights promoted by Spain (LOPD 15/1999) and the European Union EU (2000/C364/01) were followed. All patients' data were processed according to the Declaration of Helsinki (last revision 2013) and Spanish National Biomedical Research Law (14/2007, July 3).

### Sample size calculation

Since our population is finite (we know the total population), the number of individuals required to study from the total can be calculated with the formula: $n = (N \times Z\alpha^2 \times p \times q) \div (d^2 \times (N-1) + Z\alpha^2 \times p \times q)$; where N=Number of individuals; $Z\alpha=1,96$ for a level of confidence according to the standard normal distribution of 95% (Robert D Mason, Essentials of statistics. Ed Prentice Hall. 1976); p=expected probability (in this case 5%=0.05).

q=1 – p (in this case 95%=0,95); d=maximum error allowed (in this case un 5%; 0,05).

N=Given that colorectal cancer incidence in Spain is 32,240 patients, approximately 20% of them are being treated in Madrid (6448 patients) and approximately 40% of these patients are being treated in our hospital (2579 patients).

$N = (2579 \times 1.96^2 \times 0.05 \times 0.95) \div (0.05^2 \times (2579–1) + 1.96^2 \times 0.05 \times 0.95) = 71$ patients initially required. Our patient sample contains more than 71 patients.

## Colorectal cell panels

Human healthy intestinal cells (HIEC6) and colorectal adenocarcinoma (HT-29 and HCT 116 and HCT 116 derived ShControl and ShVDR) cells were cultured under standard conditions, in 5%CO2 at 37 °C with DMEM containing 25 mM glucose (unless specifically indicated) and 40 mM LiCl to mimic Wnt signalling and supplemented with 10% foetal bovine serum (FBS) and 1% penicillin-streptomycin. Cells were treated as indicated for 24 hr. HCT 116 ShControl and ShVDR cell lines were derived previously (*Larriba et al., 2011*). Cell lines used in this work are described in the table of Key resources. Cell lines were provided by the ATCC, authenticated by STR profiling and mycoplasma contamination tests by PCR are run every week in our tissue culture facilities to ensure no contamination. None of these cell lines are included in the list of misidentified cell lines https://iclac.org/databases/cross-contaminations.

## Transient transfections

cells were seeded in plates at 50% confluence for plasmid transfection, using JetPei PolyPlus reagent (Genycell Biotech), following the manufacturer's instructions. After 24 hr cells were treated as indicated for 24 hr.

For Sirt1 siRNA, cells plated in six well plates at 50% confluence were transfected with JetPRIME PolyPlus reagent (Genycell Biotech) following the manufacturer's instructions. After 2 days cells were treated 24 hr with LiCl and then another 24 hr with $1,25(OH)_2D_3$ before harvesting to analyse by western blot.

## Site directed mutagenesis

Directed mutagenesis was performed using the QuickChange Site-Directed Mutagenesis Kit (Stratagene). Oligonucleotides containing the mutation (Key Resources Table) were designed following the manufacturer's recommendations. 2.5 U of PfuTurbo DNA polymerase was added to each sample and they were amplified by performing the following cycles: 1 denaturation cycle at 95 ° C for 1 min, 18 amplification cycles of 30 s at 95 ° C, 50 s at 60 ° C, 1 min / kilobase of plasmid at 68 °C and the last extension cycle 7 min at 68°C. Once amplified, 10 U of Dnp I enzyme were added and incubated at 37 ° C for 1 hr to digest the supercoiled double-stranded DNA and eliminate the parental molecules that are not mutated.

The correct introduction of the mutation was evaluated by plasmid sequencing using the BigDye Cycle Sequencing Kit (Applied Biosystems).

## Preparation of cell extracts

### Whole cell extracts

Cells were washed with iced PBS before extract preparation and scraped in RIPA buffer (10 mM Tris HCl [pH 7.4], 5 mM EDTA, 5 mM EGTA, 1% Tryton X100, 10 mM $Na_4P_2O_7$ [pH 7.4], 10 mM NaF, 130 mM NaCl, 0.1% SDS, 0,5% Na-deoxycholate). After 5 min on ice, cells were pelleted (12,000 rpm for 5 min, 4 °C) and the supernatant was directly used as whole cell extract or frozen at –80 °C.

### Fractionated cell extracts

After washing as before, cells were scraped in hypotonic buffer (20 mM Hepes, [pH 8.0], 10 mM KCl, 0,15 mM EDTA, 0,15 mM EGTA, 0,05% NP40 and protease inhibitors) and incubated on ice for 10 min before adding 1:2 vol of sucrose buffer (50 mM Hepes [pH = 8.0], 0.25 mM EDTA, 10 mM KCl, 70% sucrose). Lysates were fractionated (5000 rpm for 5 min at 4 °C) to obtain the cytoplasmic fraction in the supernatant. Nuclear pellets were washed twice with washing buffer (20 mM Hepes [pH 8.0], 50 mM NaCl, $MgCl_2$ 1.5 mM, 0.25 mM EDTA, 0.15 mM EGTA, 25% glycerol and protease inhibitors), pelleted at 5000 rpm, 5 min at 4 °C and resuspended in nuclear extraction buffer (20 mM Hepes[pH 8.0], 450 mM NaCl, $MgCl_2$ 1.5 mM, 0,25 mM EDTA, 0.15 mM EGTA, 0,05% NP40, 25% glycerol and protease inhibitors) before centrifugation at 12,000 rpm for 5 min at 4 °C to pellet and discard cell debris. The supernatants were used as nuclear fractions.

## SIRT1 activity assay

$NAD^+$-dependent deacetylase activity was measured on nuclear extracts of HT-29 cells treated as indicated, using SIRT-Glo Assay kit (Promega), following the manufacturer's instructions. For each

reaction, 1 µg of protein was incubated with SIRT-Glo Reagent Mix at room temperature for 3 min, and the luminescence was measured using a GloMax Microplate Reader (Promega). Relative luciferase units (RLU) were calculated as fold induction relative to the corresponding control.

## Nuclear NAD⁺ abundance

The NAD/NADH-Glo Assay Kit (Promega) was used to measure $NAD^+$ abundance in HCT 116 nuclear extracts. Twenty µg of protein were diluted in 50 µl of lysis buffer (described for cell extracts) for each reaction and 25 µl of 0.4 N HCl were added before incubation at 60 °C for 15 min. This exploits the high stability of $NAD^+$ under acidic conditions, while NADH breaks down. Samples were cooled down to room temperature for 10 min and neutralized with 25 µL 0.5 M Trizma. $NAD^+$ levels were then measured using a GloMax microplate reader (Promega) and following the manufacturer's instructions.

## Immunoprecipitation

For immunoprecipitation from fractionated extracts, the hypotonic buffer was modified by adding 100 mM NaCl and 0.1% NP40. For immunocomplex formation, protein A/G-coated magnetic beads (Invitrogen) were washed 3 times with the extraction buffer before coating with the primary antibody for 2 hr at 4 °C in a rotating wheel, followed by 2 washes with the same buffer to eliminates unbound antibody and then extracts were added O/N at 4 °C in the rotating wheel. Immunocomplexes were washed twice and used for western blotting.

## Western blotting

Proteins from lysed cells or immunoprecipitates were denatured and loaded on sodium dodecyl sulfate polyacrylamide gels and then transferred to polyvinylidene difluoride membranes (Bio-Rad). After blocking with 5% (w/v) BSA or milk the membrane was incubated with the corresponding primary and secondary antibodies (Bio-Rad). The specific bands were analyzed using Thyphoon or ChemiDoc Imaging Systems (Bio-Rad).

## Immunoflorescence

Cells in cover slips were washed three times and fixed with 4% paraformaldehyde in PBS [pH 7.4] for 10 min, washed again, permeabilized (PBS [pH7.4], 0.5% Triton X-100, 0.2% BSA) for 5 min, blocked (PBS [pH7,4], 0,05% Triton X-100, 5% BSA) for 1 hr at room temperature and incubated with primary antibody over night at 4 °C. After three washes (5 min), cells were incubated with secondary antibody for 1 hr at room temperature. Slides were mounted and images were acquired using a SP5 confocal microscope (Leica) with a 63 x objective. Fluorescence intensity was quantified using Image J software. For each experiment, three different fields were evaluated per slide.

## Cell-growth curves and cell cycle

Cell growth was determined by colorimetry. For that, cells were seeded at a density of 20,000 cells per well in a Corning 12-well plate and treated as indicated for 1–6 days. After that cells were treated with 3-(4,5-dimethylthiazol-2-yl)–2,5-diphenyltetrazolium bromide (MTT) (Sigma-.Aldrich) in a 1:10 ratio in a culture medium and incubated at 37 °C for 3 hr. In living cells, MTT is reduced to formazan, which has a purple color. The medium was removed, and the formazan was resuspended in dimethylsulfoxide (DMSO), transferred to p96 plates and analyzed by a Spectra FLUOR (Tecan) at 542 nm. Viability was measured in four independent experiments and then duplicated.

For cell cycle analysis, cells were harvested by trypsinization, washed with PBS, fixed in 70% ethanol, stained with 7-AAD (Santa Cruz Biotechnology) for 10 min at 37 °C, and analyzed by flow cytometry (FACSCalibur, Becton-Dickinson). The percentages of cells at each cell cycle phase were analyzed using CXP software (Becton-Dickinson).

## RT-qPCR

Total RNA was extracted from 3 replicates of colorectal cells using TRIzol reagent (Invitrogen). Reverse transcription of 1 µg of RNA was performed according to the manufacturer's instructions Reagents and detection systems were from Applied biosystems. 18 S ribosomal RNA primers served as a nonregulated control. Relative expression was calculated using the Ct method, expressed as $2^{-\Delta\Delta Ct}$ (*Livak and Schmittgen, 2001*). The PCR efficiency was approximately 100%.

## TMA, immunohistochemistry, and quantification

Tissue microarrays (TMA) containing 92 cores of stage 2 human CRC samples, were constructed using the MTA-1 tissue arrayer (Beecher Instruments, Sun Prairie) for immunohistochemistry analysis. Each core (diameter 0.6 mm) was punched from pre-selected tumour regions in paraffin-embedded tissues. We chose central areas from the tumour, avoiding foci of necrosis. Staining was conducted in 2 µm sections. Slides were deparaffinized by incubation at 60 °C for 10 min and incubated with PT-Link (Dako, Agilent) for 20 min at 95 °C in low pH to detect VDR, SIRT1 and acetyl lysines. To block endogenous peroxidase, holders were incubated with peroxidase blocking reagent (Dako, Agilent) and then with (1:50) dilutions of antibodies: anti-VDR, anti-SIRT1 and (1:300) anti-acetyl lysine, overnight at 4 °C. All previously described antibodies presented high specificity. After that, slides were incubated for 20 min with the appropriate anti-Ig horseradish peroxidase-conjugated polymer (EnVision, Dako, Agilent). Sections were then visualized with 3,3'-diaminobenzidine (Dako, Agilent) as a chromogen for 5 min and counterstained with Harrys' Haematoxylin (Sigma-Aldrich, Merck). Photographs were taken with a stereo microscope (Leica DMi1). According to the human protein atlas (available at http://www.proteinatlas.org), a human colon tissue was used as a positive control for immunohistochemical staining to determine anti-VDR concentration and a human testis tissue for anti-SIRT1. Immunoreactivity was quantified blind with a Histoscore (H score) that considers both the intensity and percentage of cells stained for each intensity (low, medium, or high) following this algorithm (range 0–300): H score = (low%)×1 + (medium%)×2 + (high %)×3. Quantification for each patient biopsy was calculated blindly by 2 investigators (MJFA and JMU). VDR and SIRT1 showed nuclear staining, whereas acetyl lysines were nuclear and extranuclear. Clinicopathological characteristics of patients are summarized in *Table 1*.

## Bioinformatics analyses

Gene expression (RNA-seq) data from the TCGA colon adenocarcinoma cohort (COAD) was downloaded from the cBioportal for Cancer Genomics (http://www.cbioportal.org) using CGDS-R *Cerami et al., 2012*; *Gao et al., 2013* following TCGA guidelines (http://cancergenome.nih.gov/publications/publicationguidelines). SIRT1 and VDR gene expression from samples of colorectal cancer and healthy tissue adjacent to the tumour were analysed with TNMplot.com (https://tnmplot.com/analysis/) tool (*Bartha and Győrffy, 2021*). An asymptotic Spearman correlation test using original log2 expression values was used to determine the significance of the Spearman rank correlation.

## Statistical analysis

Results are presented as fold induction, mean ± SEM from three biological replicates. To determine whether calculated Hscores for protein levels were well-modelled by a normal distribution, Kolmogorov-Smirnov test was used. Association between SIRT1 and VDR protein levels was performed with Chi-square test. Comparisons between two independent groups (healthy versus tumors) were performed with Mann-Whitney U-test. For statistical association, Hscore of antigens were categorized as high- or low- expression levels using the median as a cut-off point. Tests for significance between two sample groups were performed with Student's t test and for multiple comparisons, ANOVA with Bonferroni's post-test.

**Table 1.** Clinicopathological characteristics of CRC included in the study.

**a. Patients from HCSC**

| Patients Characteristic | N (%) | |
|---|---|---|
| | Stage II | Stage IV |
| Gender | | |
| Male | 57 (60%) | 27 (48%) |
| Female | 38 (40%) | 23 (41%) |
| N/A | 0 | 6 (11%) |
| Tumor Location | | |
| Cecum | 13 (14%) | 3 (5%) |
| Right | 25 (26%) | 10 (18%) |
| Transverse | 7 (8%) | 3 (5%) |
| Left | 5 (5%) | 2 (4%) |
| Sigma | 24 (25%) | 21 (38%) |
| Rectum | 21 (21%) | 12 (21%) |
| N/A | 0 | 5 (9%) |
| Grade Primary Tumor | | |
| Well differentiated | 18 (19%) | 15 (27%) |
| Moderately differentiated | 69 (73%) | 27 (48%) |
| Poorly differentiated | 8 (8%) | 5 (9%) |
| N/A | 0 | 9 (16%) |
| Paired Liver Metastasis | | 54 (96%) |
| N/A | | 2 (4%) |
| Grade Metastasis Tumor | | |
| Well differentiated | | 12 (21%) |
| Moderately differentiated | | 29 (52%) |
| Poorly differentiated | | 4 (7%) |
| N/A | | 11 (20%) |
| Metastasis at diagnosis | | |
| Synchronous | | 35 (62%) |
| Metachronous | | 21 (38%) |
| Pt | | |
| T1 | 3 (3%) | 2 (4%) |
| T2 | 30 (32%) | 5 (9%) |
| T3 | 61 (64%) | 42 (75%) |
| T4 | 1 (1%) | 4 (7%) |

*Table 1 continued on next page*

*Table 1 continued*

**a. Patients from HCSC**

| | | |
|---|---|---|
| N/A | | 3 (5%) |
| <u>pN</u> | | |
| N0 | 95 (100%) | 27 (48%) |
| N1 | | 16 (29%) |
| N2 | | 9 (16%) |
| N3 | | 1 (2%) |
| N/A | | 3 (5%) |

**b. Patients from HUFA**

| Patients Characteristic | N (%) | | |
|---|---|---|---|
| | <u>Stage I</u> | Stage II | Stage III |
| <u>Gender</u> | | | |
| Male | 5 (31%) | 1 (6 %) | 6 (38%) |
| Female | 0 | 3 (19%) | 1 (6 %) |
| N/A | 0 | 0 | 0 |
| <u>Tumor Location</u> | | | |
| Cecum | 0 | 0 | 0 |
| Right | 2 (40%) | 1 (25%) | 2 (29%) |
| Transverse | 0 | 0 | |
| Left | 1 (20%) | 1 (25%) | |
| Sigma | 2 (40%) | 2 (50%) | 3 (43%) |
| Rectum | 0 | 0 | 2 (29%) |
| N/A | 0 | 0 | |
| <u>Grade Primary Tumor</u> | | | |
| Well differentiated | 2 (40%) | 1 (25%) | 1 (14%) |
| Moderately differentiated | 3 (60%) | 5 (50%) | 4 (57%) |
| Poorly differentiated | 0 | 1 (25%) | 1 (14%) |
| Mucinous | 0 | 0 | 1 (14%) |
| N/A | 0 | 0 | 0 |
| <u>pT</u> | | | |
| T1 | 1 (20%) | 0 | 1 (14%) |
| T2 | 1 (20%) | 0 | 0 |
| T3 | 3 (60%) | 3 (75%) | 3 (43%) |
| T4 | 0 | 1 (25%) | 3 (43%) |

*Table 1 continued on next page*

*Table 1 continued*

| b. Patients from HUFA | | | |
|---|---|---|---|
| N/A | 0 | 0 | 0 |
| <u>pN</u> | | | |
| N0 | 5 (100%) | 4 (100%) | 0 |
| N1 | 0 | 0 | 6 (86%) |
| N2 | 0 | 0 | 1 (14%) |
| N3 | 0 | 0 | 0 |
| N/A | 0 | 0 | 0 |

## Acknowledgements

Funding for this work was provided by the Agencia Estatal de Investigación (PID2019-104867RB-I00/AEI/10.13039/501100011033, RTI2018-099343-B-100 and PID2021-127645OA-I00); Instituto de Salud Carlos III (CIBERONC, CB16/12/00273 and CB16/12/00326); Comunidad de Madrid (Ayudas Atracción de Talento 2017-T1/BMD-5334 and 2021–5 A/BMD-20951; A385-DROPLET Young Reserchers R&D Project 2019 CAM-URJC; PRECICOLON-CM, P2022/BMD7212); Universidad Rey Juan Carlos (ADIPOMELM, Proyecto Puente de Investigación 2020).

We thank Lucille Banham for her valuable assistance in the preparation of the English manuscript and María Gutiérrez Salmerón for help making figures.

## Additional information

### Funding

| Funder | Grant reference number | Author |
|---|---|---|
| Agencia Estatal de Investigación | PID2019-110998RB-I00 AEI/10.13039/501100011033 | Custodia García-Jiménez |
| Instituto de Salud Carlos III | CIBERONC CB16/12/00273 | Alberto Muñoz |
| Comunidad de Madrid | PRECICOLON-CM P2022/BMD7212 | Custodia García-Jiménez |
| Comunidad de Madrid | Ayudas Atracción de Talento 2017- T1/BMD-5334 | Ana Chocarro-Calvo |
| Comunidad de Madrid | A385-DROPLET Young Researchers R&D Project 2019 | Ana Chocarro-Calvo |
| Universidad Rey Juan Carlos | ADIPOMELM | Ana Chocarro-Calvo |
| Agencia Estatal de Investigación | RTI2018-099343-B-100 | Antonio De la Vieja |
| Agencia Estatal de Investigación | PID2021-127645OA-I00 | Ana Chocarro-Calvo |
| Agencia Estatal de Investigación | PID2019-104867RB-I00 | Custodia García-Jiménez |
| Instituto de Salud Carlos III | CIBERONC CB16/12/00326 | Alberto Muñoz |
| Comunidad de Madrid | Ayudas Atracción de Talento 2021-5A/BMD-20951 | Ana Chocarro-Calvo |

| Funder | Grant reference number | Author |
|--------|------------------------|--------|
| Universidad Rey Juan Carlos | Proyecto Puente de Investigación 2020 | Ana Chocarro-Calvo |

The funders had no role in study design, data collection and interpretation, or the decision to submit the work for publication.

## Author contributions

José Manuel García-Martínez, Conceptualization, Data curation, Formal analysis, Validation, Investigation, Methodology; Ana Chocarro-Calvo, Resources, Formal analysis, Supervision, Investigation; Javier Martínez-Useros, Formal analysis, Investigation, Methodology; María Jesús Fernández-Aceñero, Formal analysis, Validation, Visualization, Methodology; M Carmen Fiuza, Data curation, Formal analysis, Validation, Investigation, Methodology; José Cáceres-Rentero, Formal analysis, Investigation, Visualization, Methodology; Antonio De la Vieja, Resources, Methodology; Antonio Barbáchano, Data curation; Alberto Muñoz, Conceptualization, Formal analysis, Supervision, Funding acquisition, Writing – review and editing; María Jesús Larriba, Supervision, Investigation, Writing – review and editing; Custodia García-Jiménez, Conceptualization, Resources, Data curation, Formal analysis, Supervision, Funding acquisition, Validation, Investigation, Visualization, Writing – original draft, Project administration, Writing – review and editing

## Author ORCIDs

Custodia García-Jiménez https://orcid.org/0000-0003-0146-4424

## Ethics

Patient samples were derived from surgical removal at Fundación Jimenez Diaz University Hospital, General and Digestive Tract Surgery Department: 95 patient samples diagnosed with stage II CRC. The Institutional Review Board (IRB) of the Fundación Jimenez Diaz Hospital, reviewed and approved the study, granting approval on December 9, 2014, act number 17/14. Samples from seven healthy individuals were obtained from 348 Hospital Clínico San Carlos (HCSC). IRB-HCSC act number 21/498-E granting approval on June 25th, 2021. Paired fresh frozen samples from tumor and nontumor adjacent tissue were collected from 18 patients at Hospital Universitario Fundación Alcorcón (HUFA). IRB-HUFA, act number 17/68 granting approval on June 8th, 2017. Colon cancer tissues were collected using protocols approved by the corresponding Ethics Committees and following the legislation. All patients gave written informed consent for the use of their biological samples for research purposes. Fundamental ethical principles and rights promoted by Spain (LOPD 15/1999) and the European Union EU (2000/C364/01) were followed. All patients' data were processed according to the Declaration of Helsinki (last revision 2013) and Spanish National Biomedical Research Law (14/2007, July 3).

Reviewer #1 (Public Review): https://doi.org/10.7554/eLife.86913.3.sa1
Reviewer #2 (Public Review): https://doi.org/10.7554/eLife.86913.3.sa2
Author Response https://doi.org/10.7554/eLife.86913.3.sa3

# Additional files

## Supplementary files

• MDAR checklist

## Data availability

All data generated or analysed during this study are included in the manuscript and supporting file uploaded at dryad: https://doi.org/10.5061/dryad.kwh70rz9d.

The following dataset was generated:

| Author(s) | Year | Dataset title | Dataset URL | Database and Identifier |
|---|---|---|---|---|
| García-Jiménez C, García-Martínez JM, Chocarro-Calvo A, Martínez-Useros J, Fernández-Aceñero MJ, Fiuza MC, Cáceres-Rentero J, de la Vieja A, Barbáchano A, Muñoz A, Larriba MJ | 2023 | Vitamin D induces SIRT1 activation through K610 deacetylation in colon cancer | http://doi.org/10.5061/dryad.kwh70rz9d | Dryad Digital Repository, 10.5061/dryad.kwh70rz9d |

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
