## [Editor Report · eLife assessment]

This study demonstrates that vitamin D-bound VDR increased the expression of SIRT1 and that vitamin D-bound VDR interacts with SIRT1 to cause auto-deacetylation on Lys610 and activation of SIRT1 catalytic activity. This is an **important** finding that is relevant to the actions of VDR on colorectal cancer. The data presented to support the presented conclusion are **convincing**.

---

## [Referee Report · Reviewer #1 (Public Review)]

This study demonstrates that vitamin D-bound VDR increased the expression of SIRT1 and that vitamin D-bound VDR interacts with SIRT1 to cause auto-deacetylation on Lys610 and activation of SIRT1 catalytic activity. This is an important finding that is relevant to the actions of VDR on colorectal cancer. The data presented to support the presented conclusion is convincing.

A strength of the study is that it is focused on a narrow group of conclusions.

The major weakness of the study is that the site of SIRT1 regulatory lysine acetylation is defined by mutational analysis rather than by direct biochemical analysis. This issue is partially mitigated by previous reports of K610 acetylation using mass spec (https://www.phosphosite.org/proteinAction.action?id=5946&showAllSites=true). However, Fig. 4E is reassuring because it shows that the apparent acetylation of the K610 mutant SIRT1 appears to be lower than WT SIRT1

A second weakness of the study relates to the use of shRNA-mediated knockdown of VDR for some studies in which a previously reported cell line was employed. The analysis presented would be more compelling if similar data was obtained using more than one shRNA. Similarly, only a single siRNA for SIRT1 is presented in Table 1.

A third weakness of the study is that the conclusion that the VDR interaction with SIRT1 is the cause of auto-deacetylation rather than an associated event mediated by another mechanism would be more strongly supported by mutational analysis of SIRT1 and VDR residues required for the binding interaction. Will VDR increase SIRT1 activity when mutations are introduced to block the interaction? While the finding that catalytically inactive SIRT1 does not interact with VDR is helpful, this does not address the role of the binding surface.

A fourth weakness of the study is that it would be improved by testing the proposed hypothesis through in vitro reconstitution with purified proteins. Does VDR cause auto-deacetylation and activation of Sirt1 in vitro?

---

## [Referee Report · Reviewer #2 (Public Review)]

The authors provide a comprehensive analysis of vitamin D-mediated signaling through VDR, SIRT1, and Ace H3K9. They specifically emphasize the significance of K610 in SIRT1 within this signaling pathway. The article effectively presents a convincing and straightforward argument. The experiments conducted are meticulously executed, and the statistical analysis is sound. The inclusion of complex biochemistry details adequately covers the topic at hand. These findings hold great relevance to both normal and pathological physiology across different cell lineages, making them of considerable interest.

---

## [Author Response]

The following is the authors’ response to the original reviews.

We thank the reviewers and appreciate their recommendations to improve this work.

**Reviewer 1:**

Reviewer 1 recognizes that ‘This is an important finding that is relevant to the actions of VDR on colorectal cancer. The data presented to support the presented conclusion is convincing’.

Reviewer 1 identifies as a major weakness ‘that the site of SIRT1 regulatory lysine acetylation is defined by mutational analysis rather than by direct biochemical analysis.

However, as the reviewer mentions “previous reports of K610 acetylation using mass spec (https://www.phosphosite.org/proteinAction.action?id=5946&showAllSites=true), and the absence of SIRT1 mutant K610R in the immunoprecipitates using anti-acetylated lysine antibodies presented in Fig. 4E clearly overcome this weakness”.

In addition, overall SIRT1 acetylation is reduced by vitamin D and by the specific SIRT1 activator SRT1720 as shown by decreased SIRT1 in the anti-acetyl-lysine immunoprecipates, (Fig. 4A and B).The second weakness identified by Reviewer 1 concerns “the use of only one shRNA to deplete VDR in CRC cells.”

We have made efforts to demonstrate that the results are specific, though we do not have results with alternative shRNAs for a variety of reasons. To mitigate this issue, we have compared two colon cancer cells originating from the same patient which differ in the presence/absence of VDR. SW480, derive from the primary tumor and express VDR, whereas SW620 cells were derived from a lung metastasis and lack VDR. Similar, to the comparison of HCT116 with shVDR HCT116 cells presented in this study, VD induced SIRT1 levels in SW480 in contrast to a lack of induction in SW620, as shown in Author response image 1. This result provides support for the specificity of the shVDR.

**Author response image 1. sa3fig1:** Vitamin D requires the presence of VDR to increase SIRT1 protein levels. SW480 and SW620 cell lines derive from the same patient, from primary tumor and lung metastasis respectively and differ in their VDR content. 1α,25-dihydroxyvitamin D3 (1,25(OH)2D3) was added at 100 nM for 24 h. Representative western-blot, where TBP was used as a loading control, of four biological replicates. Statistical analysis by ANOVA and values represent mean ± SEM; *p<0.05; *** p<0.001.

The referee noticed the inclusion of an siRNA for SIRT1 in Table 1. We apologize for that, since this is an error, and no results are presented in this study with SIRT1 depletion. Table 1 has been modified accordingly.

Concerning the third and fourth weaknesses that Reviewer 1 identifies, we agree that mapping the interacting domains in both VDR and SIRT1 and in vitro reconstitution would improve the present study. However, we believe that these would constitute long-term studies that themselves are not strictly necessary at this stage. Consequently, we favor the publication of the present body of work. In vitro reconstitution of the present work and the putative relevance of the proposed mechanism of vitamin D action via SIRT1 on types of cancer other than colon (eg breast etc), are certainly very interesting and warrant further investigation.

**Reviewer 2:**

This reviewer acknowledges that “…this study provides very interesting and solid information on the link between vitamin D and colorectal cancer. It is likely that this study will provide insight into the importance of vitamin D in other types of cancer. It may also lead to new therapeutic strategies for specific cases. This article is convincing, although the authors can improve their study as outlined…”

We acknowledge the proposed changes and recommendations, and have changed the text and Figures as suggested the by Reviewer as follows:

Figure 1

Figure 1E and F: the cell lines used were described in the figure legend, but we agree that including the name in the figure brings more clarity and these are now added.

Figure 1G: the statistical analysis was for all panels of Figure 1 as described in the Figure legend (lines 731-32), We have amended the original omission of panels 1G and 1H. In panel G, * represents statistical analysis by ANOVA (comparing the four groups) whereas # was the analysis by Students t test (comparing the two indicated groups), where * or #p<0.05. We hope to have clarified this point now.

Figure 2

Figure 2C: We showed originally the SIRT1/VDR interaction by immunoprecipitation of VDR and detection of SIRT1 in immunoprecipitates. We also showed immunoprecipitation of exogenously expressed Myc-SIRT1 (WT or mutants) and detection of VDR in immunoprecipitates (Figure 4F). The reviewer requests that we perform the inverse IP for endogenous SIRT1, that is immunoprecipitate SIRT1 and detect VDR in the immunoprecipates, which we now supply for the reviewer in Author response image 2.

**Author response image 2. sa3fig2:** Immunoprecipitation of endogenous SIRT1 to show interaction with VDR. 1α,25-dihydroxyvitamin D3 (1,25(OH)2D3) was added at 100 nM for 24 h. Representative western-blots, where TBP was used as a loading control.

Figure 3

Figure 3D: ‘The authors should indicate the color of the different stainings’. Immunostainings have been revealed with DAB (diaminebenzidine); thus, positiveness is highlighted by light or dark brown according to their low or high protein expression. Counterstaining has been performed with hematoxilin, which stains nuclei in dark blue and cytoplasm in light blue.

Do the authors mean that the secondary antibody marks in brown/red? If so, these results are inconsistent with the text considering that hematoxylin was used for non-tumor tissue. This part needs to be clarified.

We thank the Reviewer for asking us to clarify this issue. Neither the primary nor anti-Ig horseradish peroxidase-conjugated secondary antibodies presented positiveness resulting from these antibodies individually. Therefore, secondary antibody does not mark in any color. Hematoxylin has been used as counterstaining for both non-tumor as well as for tumor tissues.

What about the level of FOXO3A in these tissues/tumors?

We did not prove the tumor sections for specific SIRT1 substrates such as FoxO3A since their levels may not entirely depend on SIRT1 specific deacetylation.

What is the level of 1,25(OH)2D3 in these patients?

We agree with this referee that this information would be very useful, but unfortunately, we do not have data on vitamin D levels for these patients since they were not specifically recruited for this study and vitamin D levels are not routinely measured.

Figure 3D, the following information is missing: "A detailed amplification is shown in the lower left of each micrograph."

We decided not to include the amplification in micrographs because the aim of the manuscript is focused on protein levels, not localization and including the amplification was more confusing than enlightening. This has been amended now in the text.

Figure 3E, it says p=0.325, in the legend p<0.01, and in the text there is a trend. Which is the correct version?

We really apologize for this misunderstanding. As stated in the Figure, p=0.325 and therefore it does not reach statistical significance. We have amended the main text and figure legend to report that differences between SIRT1 expression levels of healthy and cancer human colon samples are not statistically significant.

Figure 4Figure 4F. The quality of the presented blots is not optimal. It needs to be improved. In addition, the number of independent biological experiments is not indicated.

We have substituted the representative western-blot and included statistical analysis of four independent biological replicates. Since 4F is now a bigger panel, it has required a slight reorganization of the whole Figure, but the rest of panels remain with the originals. Now we indicate in the figure legend that at least three independent biological replicas were analyzed. In addition, we supply below the four experiments for the reviewer in Author response image 3.

**Author response image 3. sa3fig3:** Immunoprecipitation of exogenous myc-tagged SIRT1 to show interaction with VDR of wild type (WT) or mutants. 1α,25-dihydroxyvitamin D3 (1,25(OH)2D3) was added at 100 nM for 24 h. FT: Flow Through. TBP as a loading control.

Regarding the last general comment concerning the number of independent experiments performed, this is indicated in the Figure legends (lines 732-36, 757-58, 82324, 840-41). All the in vitro experiments were performed at least as three independent experiments and not by repeating a western blot. A representative western blot is shown, and the statistical analysis corresponds to the analysis of the three biological replicates. For experiments with patient samples, the number of patients appears clearly indicated in the corresponding panel.